# The Effect of Tai Chi Chuan on Negative Emotions in Non-Clinical Populations: A Meta-Analysis and Systematic Review

**DOI:** 10.3390/ijerph16173033

**Published:** 2019-08-21

**Authors:** Shu Zhang, Liye Zou, Li-Zhen Chen, Ying Yao, Paul D. Loprinzi, Parco M. Siu, Gao-Xia Wei

**Affiliations:** 1Key Laboratory of Behavioral Science, Institute of Psychology, Chinese Academy of Sciences, Beijing 100101, China; 2Key Laboratory of Mental Health, Institute of Psychology, Chinese Academy of Sciences, Beijing 100101, China; 3Department of Psychology, University of Chinese Academy of Sciences, Beijing 100049, China; 4Exercise and Mental Health Laboratory, School of Psychology, Shenzhen University, Shenzhen 518061, China; 5Exercise Psychology Laboratory, Department of Health, Exercise Science and Recreation Management, University of Mississippi, 215 Turner Center, Oxford, MS 38677, USA; 6Division of Kinesiology, School of Public Health, Li Ka Shing Faculty of Medicine, The University of Hong Kong, Pokfulam, Hong Kong 999077, China

**Keywords:** Tai Chi Chuan, prevention, depression, anxiety, exercise

## Abstract

Tai Chi Chuan (TCC) as a typical mind-body practice has been investigated for its preventive role on negative emotions and has demonstrated its efficacy in healthy populations. However, the results are not consistent. We performed a meta-analysis and systematically evaluated the effect of TCC on selected negative emotions (i.e., anxiety and depression). Fourteen experimental studies from three English-and two Chinese-language databases were evaluated. The results showed that the positive effects of TCC on negative emotions were moderately to largely significant. In addition, although TCC significantly improved negative emotions in both the young adults and the older adults, old adults benefited more from TCC than young adults. These findings suggest that TCC is a worthy complementary non-pharmacological resource towards depression and anxiety and, thus, has great implications for the public health domain.

## 1. Introduction

The World Health Organization reported that about 8% of the world’s population suffers from depression and anxiety, which are one of the leading causes of disability and mortality across the lifespan [1]. Emotional disorders not only exert significant impact on individual health and quality of life, but place enormous burden on family and society as well. However, more than 80% of people do not seek treatment in developing countries and only 35–50% in high-income countries receive minimally adequate treatment [2] because of barriers such as high cost, limited valid treatments and diagnosis difficulty. Moreover, the recognized treatments for emotional disorders, namely, pharmacological therapy and psychotherapy, have been criticized for inconsistent clinical outcomes and side effects [3,4]. Given the current limitations in effectiveness of treatment modalities for emotional disorders, the WHO suggested that the most suitable method for reducing the burden caused by these disorders is prevention [5]. Hence, prevention (instead of intervention) among the non-clinical population with emotional symptoms is of great importance [6].

There is a wide range of evidence-based preventive strategies available, which have been found to reduce risk factors and strengthen protective factors relating to emotional disorders. Among these preventive approaches, exercise intervention (e.g., running, brisk walking, Yoga, and Tai Chi Chuan) (TCC) could provide both physical and psychological benefits [7]. Recently, accumulating evidence demonstrates a preventive effect of physical exercise on decreasing negative emotions in non-clinical populations. In a study involving 16,483 university undergraduates, self-reported exercise was correlated with lower depression, after controlling for age and sex [8]. Similarly, longitudinal studies among non-clinical populations indicates that physical exercise is an efficient tool to alleviate depression and decrease anxiety [9,10]. In addition to physical exercise, those alternative interventions, including mind-body practice such as TCC, Yoga, Qigong as well as mindfulness, have been widely investigated [11].

TCC is one of the Chinese traditional mind-body exercises originated from Taoism philosophy (peaceful/harmonious and holistic principle) with hundreds of years of history [12]. Notably, it seeks for the state of being aware of the permanent, unchanging center of your being as well as mind emptiness by sequence practice and slow movements, which is rooted rom Taoist neigong systems. It emphasizes the coordination of mind and body as well as simultaneous focus on relaxation and concentration. Moreover, TCC movement is relatively gentle and slow, and it does not need a special field or equipment; thus, it has been widely spread in Western countries as an alternative medicine to improve mental and physical health [13,14,15] among individuals with a metabolic syndrome [16], neurological disorder [17], cardiovascular disease [18] or fibromyalgia [19]. A growing number of clinical studies have also shown that TCC can decrease depression [20] and reduce anxiety [21] in patients with emotional disorders. Regarding the significance of prevention towards emotional disorders, it is important to investigate the potential value of TCC in non-clinical populations. Emergent studies have explored the effect of varied TCC styles on alleviating negative emotions in college students, adults and older adults; however, these studies obtained inconsistent findings. Therefore, the effect that TCC has on negative emotions in non-clinical populations remains largely unknown. Furthermore, due to different study designs, varied outcome measurements and multiple practice protocols, the findings from these studies remain inconclusive. There is no systematic study that has evaluated the effect of TCC on negative emotions in non-clinical populations. In this study, we aimed to conduct a comprehensive review of the effect that TCC has on depression and anxiety.

## 2. Materials and Methods

### 2.1. Literature Search

The Preferred Reporting Item for Systematic Reviews and Meta-Analysis guideline was employed for the procedures of this systematic review [22]. Three English-language (Pubmed, Web of Science and EBSCO) and two Chinese-language databases (China National Knowledge Infrastructure and Wanfang) were searched from inception to February 2019. To obtain a maximum of relevant studies, we used two groups of keywords: (1) “mind-body exercise” OR “mindful exercise” OR “meditative movement” OR “mindfulness-based exercise” OR “Tai Chi” OR “Taiji” OR “mind-body practice” OR “mind-body intervention”; (2) “affective” OR “mood” OR “emotion” OR “depression” OR “anxiety”. Additionally, we manually searched relevant studies from reference lists of the initially retrieved articles and reviews.

### 2.2. Inclusion Criteria and Study Selection

Studies were only included in this review if they met the following inclusionary criteria: (1) experimental studies published in peer-review journals; (2) involving participants with no physical and/or mental disorder; (3) TCC as an exercise intervention program; (4) at least one outcome measure (depression and/or anxiety) associated with negative emotion; (5) obtainable quantitative data for effect size calculation. Studies that did not meet the above-mentioned inclusion criteria were excluded (e.g., single-group study with pre-test/post-test design). All retrieved records were initially screened by three independent review authors (S.Z., L.Z.C. and Y.Y.). Further evaluation on uncertain records was conducted by a fourth review author (G.X.W.).

### 2.3. Methodological Quality Assessment of Included Studies

We used the Physiotherapy Evidence Database (PEDro) Scale [23] to assess methodological quality of all included studies. It consists of 11 evaluation items as follow: eligibility criteria, random allocation, concealed allocation, baseline equivalence, participants blinding, instructor blinding, assessor blinding, retention rate ≥85%, intention-to-treat analysis for missing data, between-group statistical comparison and point measure/measure of variability ≥ one key outcome. Given that the blinding of participants and instructors is impossible during TCC intervention, these two items (instructors blinding and subjects blinding) were removed from the scale. The evaluation was completed by three independent reviewers who examined each individual item to objectively evaluate risk of bias (e.g., selection, performance, detection or attention) across trails, but did not summarize methodological quality of each trial according to its sum score. The detailed information on quality assessment is summarized in Table 1 and Table 2.

### 2.4. Data Extraction and Analysis

Two review authors independently extracted descriptive information related to study characteristics: (1) participants (age, ratio of female and sample size); (2) TCC intervention protocol (weekly dosage, total time and qualified instructor); (3) outcomes (e.g., depression and/or anxiety) and instruments. To investigate the beneficial effects of TCC on negative emotions, we also extracted quantitative data (depression and/or anxiety).

If a study included one TCC group and two control groups (active and non-active control conditions), we only selected the non-active control-like wait-list to compare with the TCC intervention. In addition, if there were two or more active control groups (aerobic exercise, Yoga, mindfulness), we only selected the non-mindfulness-based intervention since TCC has shared similar components with Yoga and mindfulness. If two instruments measured depression, we selected the most frequently used instrument across the included studies.

To calculate pooled effect size (Standardized mean difference: SMD, negligible effect: 0–0.19; small effect: 0.2–0.49; moderate effect: 0.5–0.79; and large effect: >0.8 [9]), we used the Comprehensive Meta-Analysis Software (Bio. Stat. Inc., Englewood, NJ, USA) based on the number of participants of each group and its quantitative data (mean and standard deviation) at baseline and post-intervention. The random-effect model was selected along with 95% confidence interval (95% CI). I-squared was used to determine heterogeneity (small = 25%, moderate = 50% and large = 75%) across the selected studies. Finally, the funnel plot and Egger’s regression intercept test were employed to determine whether publication bias existed. Furthermore, we performed subgroup analysis for age range: (1) adults aged between 18 and 65 and (2) older adults aged 65 and above. Additionally, considering the differences of study design, we conducted a subgroup analysis based on study design.

## 3. Results

### 3.1. Study Selection

Both electronic and manual searches resulted in 3689 records in total. After removing the duplication and irrelevant articles, 353 full-text articles were assessed according to the pre-determined inclusion criteria, leading to a final number of 14 eligible experimental studies, including two Chinese articles. The process of study selection is shown in detail in Figure 1.

### 3.2. Study Characteristics

As is depicted in the Table 1 and Table 2, the evaluated studies were published from inception to December 2018. Six studies were conducted in China, three in the USA, two in Australia, two in Switzerland and one in Spain. The total sample size was 1285 (TCC = 645; Control = 640), with the percentage of female participants ranging from 46.67 to 95%. The mean age of each study varied from 14.79 to 80.5 years old. Both long-term intervention studies (*n* = 12) and short-term intervention studies (*n* = 2) were included in this review. For long-term intervention studies, the intervention duration lasted from 12 weeks to 18 months. Furthermore, 71.43% (10 studies) involved TCC supervisors during the intervention.

### 3.3. Study Quality Assessment

Table 3 displays the study quality of all evaluated studies. To calculate the PEDro score, if the item was not mentioned in the paper, we signed a 0 for this item in the PEDro table. In detail, one study scored 4, three studies scored 5, six studies scored 6, two studies scored 7 and two studies scored 8. The selected studies indicated fair-to-high (sum scores ranged from 4 to 8) study quality, with the mean quality score was 6.07. Furthermore, all selected studies met item 7–9 of the PEDro scale and four of selected studies did not clarify their specified criteria for participants.

### 3.4. Effects of TCC Intervention on Negative Emotion

Firstly, all selected studies were entered into the model to analyze the effects of TCC on both negative emotions (depression and anxiety). Conservatively, if a study contained more than one negative emotion, the minimum change score of the negative emotion was selected to be analyzed. The asymmetrical funnel plot was presented (Egger’s regression intercept = −5.216, *p* = 0.144). According to the funnel plot and Egger’s regression intercept, two outliers were excluded (Sattin et al., 2005 [15] and Zheng et al., 2017 [24]; Egger’s regression intercept = 0.278, *p* = 0.859). The final funnel plot is depicted in Figure 2. This meta-analysis with random model indicated a significant benefit of TCC on negative emotions, as compared to control groups (SMD = −0.500, 95% CI: −0.677 to −0.541, I^2^ = 44.835%, *p* = 0.000, Figure 3).

### 3.5. Effect of TCC Intervention on Anxiety

Nine studies (10 paired comparisons) investigated the effects of TCC on anxiety, measured by different instruments (The 80-item PHCSCS, State-trait Anxiety Inventory-state, Hamilton Depression Scale, Depression Anxiety Stress Scales, Unpleasant TESI emotion, Spielberg Anxiety Scale, SCL-90 and Self-rating Anxiety Scale). Two outliers was excluded (Zhang et al., 2014 [25] and Zheng et al., 2017 [24]) in accordance with the asymmetrical funnel plot (before removing outliers: Egger’s regression intercept = −5.956, *p* = 0.036; after removing outliers: Egger’s regression intercept = 2.391, *p* = 0.056). The final funnel plot is depicted in Figure 4. This meta-analysis with random model indicated TCC significantly decreased anxiety, as compared to control groups (SMD = −0.561, 95% CI: −0.714 to −0.408, I^2^ = 0.000%, *p* = 0.000, Figure 5).

### 3.6. Effect of TCC Intervention on Depression

Additionally, we meta-analyzed the data from nine studies on depression. This outcome was measured by different instruments, with greater scores indicating worse depression. An asymmetric funnel plot was visually observed and presented two outlying studies (Sattin et al., 2005 [15]; Egger’s regression intercept = −6.349, *p* = 0.418). After removing outliers, the funnel plot is symmetrically presented in Figure 6 (Egger’s regression intercept = −1.451, *p* = 0.584). The random model analysis indicated a significant positive effect of TCC on depression (SMD = −0.495, 95% CI: −0.762 to −0.229, I^2^ = 59.790%, *p* = 0.000, Figure 7).

### 3.7. Moderator Analysis for Age

Although most TCC studies were focused on older populations in view of its features of slow movements with moderate intensity, TCC has been employed in all-aged populations. Among the 14 studies, we found that TCC was also practiced among teenager and younger adults. Therefore, we meta-analyzed moderation analyses for age after removing the only one teenager focused study. Effect sizes were obtained from the selected 10 studies after removing outliers, which investigated the effect in adults (*N* = 7) and older adults (*N* = 5). A significant age moderator effect was found using the random-effect model: Q = 4.970, df = 1, *p* = 0.026 (adults: SMD = −0.330, 95% CI: −0.524 to −0.137, I^2^ = 0.000%, *p* = 0.001; older adults: SMD = −0.701, 95% CI: −0.946 to −0.439, I^2^ = 49.838%, *p* = 0.000, Figure 8).

### 3.8. Moderator Analysis for Experimental Design

Studies included both RCT and non-RCT designs. There was no significant moderator effect of study design with the random-effect model (Q = 0.197, df = 1, *p* = 0.657; Figure 9).

## 4. Discussion

Although the positive effect of TCC on negative emotions (anxiety and depression) have been widely demonstrated in many clinical studies [26,27,28,29], the interest on its preventive effects among non-clinical populations have only emerged in recent years. In this study, a systematic meta-analysis was comprehensively conducted to investigate the effects of TCC on negative emotions (anxiety and depression) among apparently healthy individuals, which showed that the effect of TCC practice on anxiety and depression reached moderate to large effect sizes. Additionally, our meta-analysis also suggests that TCC is an appropriate practice to prevent depression and anxiety among individuals across the adult lifespan.

The significant effects of TCC on depression and anxiety might be attributed to its meditative components. Several studies support the meditative effect of TCC, which shows that TCC practitioners have a high level of non-reactivity to inner experiences [11,30]. TCC, a traditional Chinese martial art, involving mindfulness-based slow and continuous with highly choreographed movements [31], emphasizes relaxation of musculoskeletal and the empty state of mind [32]. It is plausible that TCC reduces negative emotions by enhancing attentional control, improving emotion regulation and altering self-awareness [33] during the meditative movements. Such improvements in meditative level induced by TCC practice may result in decreased depression and anxiety [10,32,34,35,36]. Hence, the mindfulness-based movements might play an importance role in decreasing the feeling of anxiety [28,37] and depression [38,39].

On the other hand, abdominal breathing during TCC practice is likely another crucial component to decrease depression and anxiety. Abdominal breathing is usually conducted by contracting the diaphragm and expanding the abdomen. During TCC practice, the practitioners perform a series of slow movements accompanied by deep breathing. Such breathing style could help the body reach relaxation and exert a positive impact on negative emotions. The potential mechanism might be related to optimized function of the autonomic nervous system, especially altered heart rate variability (HRV). Wei et al. (2015) detected that TCC experts showed improved HRV compared to healthy controls [40]. Additionally, it was observed that HRV was significantly increased after practice among TCC individuals with depression [38]. The enhanced HRV is also a strong protective factor of cardiovascular diseases (CVD), which suggests that TCC may be an important alternative medicine for the prevention and treatment of CVD [41]. Notably, HRV is used as a reliable index of autonomic nervous modulation as well as an objective measurement of emotional regulation capacity [42,43]. Therefore, improved HRV induced by TCC practice indicates better balance between the sympathetic nervous system and parasympathetic nervous system, which leads to better emotional regulation [44]. TCC practice might alter HRV by increasing the breathing amplitude and hence enhance the capability of emotional regulation.

One mechanism accounting for the change of negative emotions might due to exercise-induced neurotrophic factor production. It is well-documented from human and animal studies that aerobic exercise produces a benefit on brain health and anxiety/depressive reduction [45]. A neurobiological mechanism is that physical exercise increases the production of insulin-like growth factor (IGF-1) and brain-derived neurotrophic factor (BDNF) [46]. IGF-1 is a critical longevity related biomarker, which plays a prominent role in the development of the central nervous system [47,48] and cell proliferation [49]. Increasing (IGF-1) is associated with improved mood, anxiety status [50] and depression [51]. Participants engaging in exercise have observed increases in IGF-1, improved positive feelings and decreased anxiety [47,52]. Brain-derived neurotrophic factor (BDNF) is another critical neurotrophic factor for neural plasticity and depressive-like behaviors [53,54,55]. Patients with depression show lower serum BDNF than healthy controls [56]. After six months of TCC practice, an upregulation of BDNF were detected among older adults with amnestic mild cognitive impairment (a-MCI) [57]. Similarly, studies on other mind-body exercise also have shown a positive effect of TCC on serum BDNF among depressed patients [58,59] and healthy people [60]. A randomized one-year aerobic exercise program for healthy older adults observed that exercise increased serum levels of BDNF and IGF-1 [61], which are protective neurobiological markers for emotional disorders [62,63].

In this study, we also performed a subgroup analysis to investigate the moderator effect of age. We found a significant moderation effect of age, which depicted a better negative emotion beneficial effect among older adults than younger adults. However, despite this moderation effect, our findings demonstrate TCC benefits on negative emotions occur in both younger and older adults. In view of its slow movement with mild to moderate intensity, TCC is believed to be mostly suitable for the elderly. For instance, it could reduce anxiety among older adults with anxiety disorder [64,65] and depression [38,66]. Moreover, since it originates from Taoism, TCC practice stresses the harmony of mind and body as well as the simultaneous state of relaxation and concentration. Currently, clinical studies (e.g., attention deficit hyperactivity disorder, asthma) also found beneficial effects on emotion among adolescence [26,27] and children [28,67], as well as young adults [29,68]. TCC contains multiple components including mindfulness, deep breathing and aerobic exercise, which contribute together to its overall effect of negative emotions towards all age populations. Given that there was only one adolescence study in the meta-analysis, we excluded it in the subgroup analysis. Thus, our results included studies from only younger adults and older adults. Previous studies demonstrate that the onset of the majority of emotional disorders occurs during adolescences [69]. Thus, future research should evaluate the effect of TCC practice on emotional disorders among adolescents.

Moreover, a subgroup analysis comparing the difference between RCT studies and non-RCT studies was conducted. Although the results did not show a significant difference between these two study designs, there were only a few RCTs. Given the powerful design of an RCT, future studies on this topic, when feasible, should implement an RCT.

## 5. Limitations

When interpreting the findings of this meta-analysis, it is important to consider that even though all of the included studies used TCC as the intervention skill, the intervention involved included both single and mixed forms combining TCC with other mind-body practices. This may render difficulty in revealing the real effect of mono TCC on emotional outcomes. Moreover, these included studies measured outcomes with different scales, which might lead to diversity of study results, as well as heterogeneity in effect size. However, the findings still support the predictions that TCC practice can reduce negative emotions in non-clinical populations.

## 6. Conclusions

This systematic review, based on existing literature, both in English and Chinese, showed that TCC practice is moderately to largely effective in improving negative emotions, decreasing depression and reducing anxiety in apparently healthy individuals. Our findings suggest that TCC could be considered a worthy complementary non-pharmacological resource for depression and anxiety, which has great implications for the prevention of emotional disorders and mental health promotion. To reach a firm conclusion, more large-scale studies with robust experimental design like randomized controlled trial, especially in young adults are needed, along with follow-up assessment.

## Figures and Tables

**Figure 1 ijerph-16-03033-f001:**
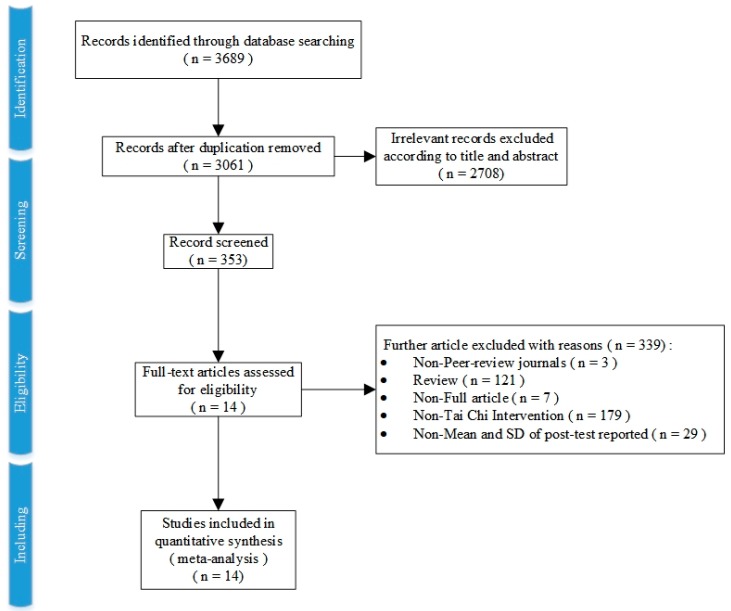
The detailed process of trial selection.

**Figure 2 ijerph-16-03033-f002:**
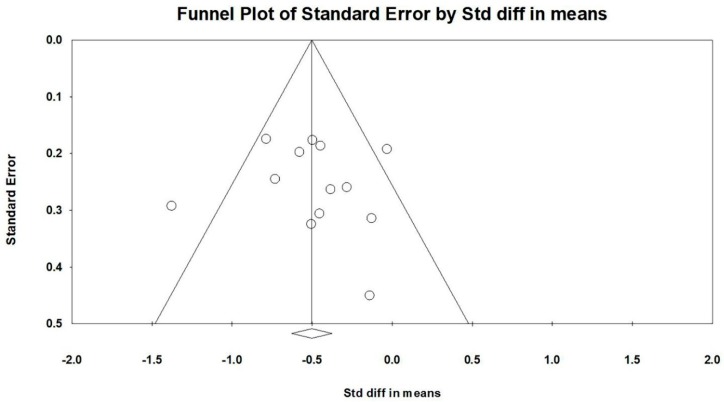
Funnel plot for negative emotions after removing outliers.

**Figure 3 ijerph-16-03033-f003:**
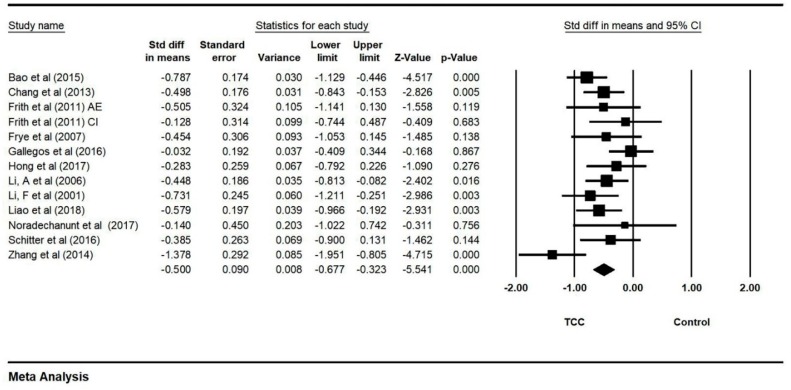
Forest plot for negative emotions (CI: circuit training; AE: aerobic exercise).

**Figure 4 ijerph-16-03033-f004:**
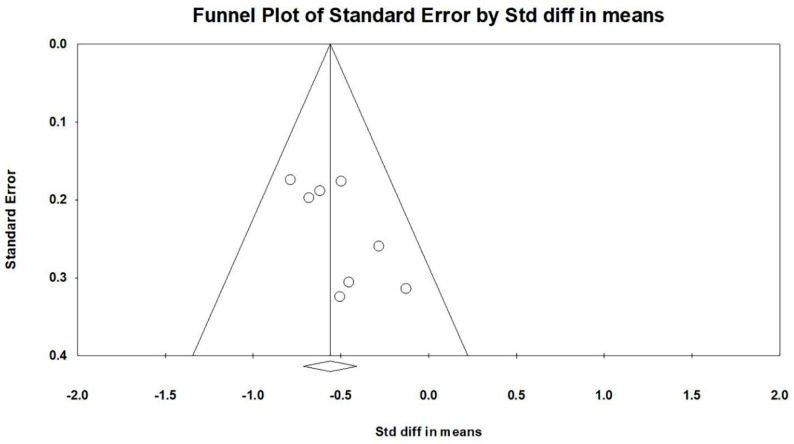
Funnel plot for anxiety after removing outliers.

**Figure 5 ijerph-16-03033-f005:**
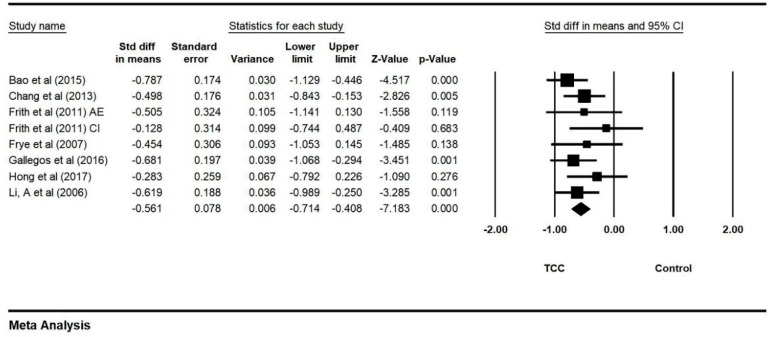
Forest plot for anxiety (CI: circuit training; AE: aerobic exercise).

**Figure 6 ijerph-16-03033-f006:**
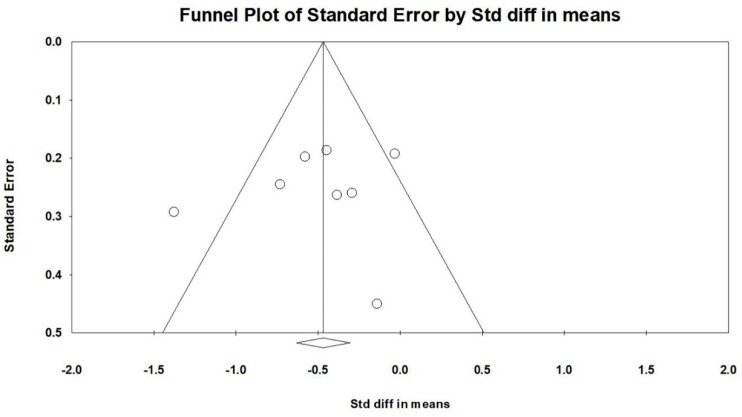
Funnel plot for depression after removing outliers.

**Figure 7 ijerph-16-03033-f007:**
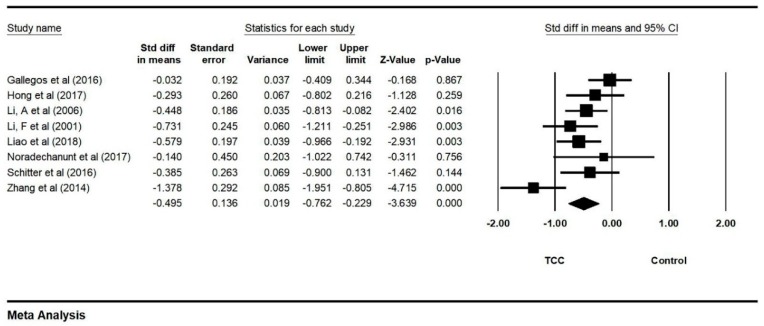
Forest plot for depression.

**Figure 8 ijerph-16-03033-f008:**
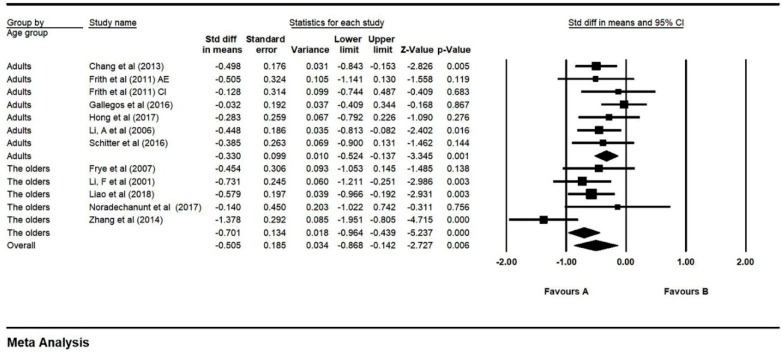
Forest plot for moderator (age) analysis (CI: circuit training; AE: aerobic exercise).

**Figure 9 ijerph-16-03033-f009:**
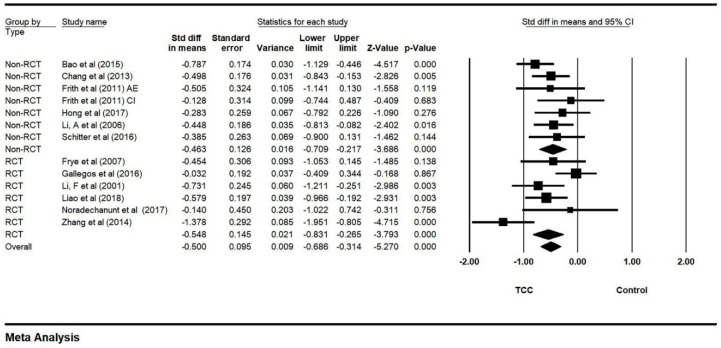
Forest plot for moderator (experimental design) analysis (CI: circuit training; AE: aerobic exercise).

**Table 1 ijerph-16-03033-t001:** The experimental characteristics of all selected trials.

Studies	Study Area	Study Design	Sample Size (N)	Female (%)	Mean Age (Years)
Bao et al. 2015	China, Mainland	PPCGD ^1^	TCC ^2^: 73; C ^3^: 69	TCC: 52.1; C: 52.2	TCC: 14.8; C: 14.7
Chang et al. 2013	China, Taipei	QED ^4^	TCC: 64; C: 69	TCC: 53.1; C: 65.2	TCC: 56.5; C: 62.3
Frith et al. 2011	Australia	Field study	TCC: 29; C1 (CI ^5^): 30; C2 (AE ^6^): 34	TCC: 62.1; C1: 53.3; C2: 50	TCC: 37.2; C1: 30; C2: 27.7
Frye et al. 2007	USA	RCT ^7^	TCC: 23; LIE ^8^: 28; C: 21	Total: 54	Total: 69.2
Gallegos et al. 2016	Spain	RCT	TCC: 68; Y ^9^: 85; M ^10^: 84; C: 45	Total: 54.6	Total: 20.3
Hong et al. 2017	China, Mainland	PPCGD	TCC: 60; C: 60	TCC: 50; C: 50	NR
Li, A et al. 2006	China, Mainland	PPCGD	TCC: 59; C: 59	Total: 47.5	NR
Li, F et al. 2001	USA	RCT	TCC: 40; C: 32	Total: 75	Total: 73.2
Liao et al. 2018	China, Mainland	RCT	TCC: 55; C: 52	TCC: 65.5; C:57.7	TCC: 71.84; C: 71.8
Noradechanunt et al. 2017	Australia	RCT	TCC: 9; Y: 11; C: 10	TCC: 69.2; Y: 76.9; C: 76.9	TCC: 67.2; Y: 67.6; C: 65.2
Sattin et al. 2005	USA	RCT	TCC: 92; WE ^11^: 82	TCC: 95; WE: 94	TCC: 80.4; WE: 80.5
Schitter et al. 2016	Switzerland	PPCGD	TCC: 28; C: 31	Total: 66.7	Total: 35.5
Zhang et al. 2014	China, Mainland	Five-arm RCT	TCC: 28; Swim ^12^: 29; Run ^13^: 27; SD ^14^: 30; C: 30	TCC: 53.6; C: 46.7	TCC: 65.5; C: 64.1
Zheng et al. 2017	Australia	Three-arm RCT	TCC: 17; Ex ^15^: 17; C: 16	TCC: 64.7; Ex: 82.4; C: 87.5	TCC: 35.4; Ex: 32; C: 34.6

^1^ PPCGD: Pre and post-test control group design; ^2^ TCC: Taichi Chuan; ^3^ C: Control; ^4^ QED: Quasi experimental design; ^5^ CI: circuit training; ^6^ AE: aerobic exercise; ^7^ RCT: random control trial; ^8^ LIE: low impact exercise; ^9^ Y: Yoga; ^10^ M: Mindfulness; ^11^ WE: Wellness Education; ^12^ Swim: swimming; ^13^ Run: running; ^14^ SD: Square Dancing; ^15^ Ex: Exercise; NR: not reported.

**Table 2 ijerph-16-03033-t002:** The intervention characteristics of all selected trials.

Studies	TCC Style	Weekly Dosage	Duration	Supervisor (Y or N)	Emotion Outcomes
Bao et al. 2015	Yang style	60 min × 5	1 year	Y ^1^	PHCSCS ^3^
Chang et al. 2013	Cheng Style	60 min × 3	12 weeks	Y	BAI ^4^
Frith et al. 2011	N/A ^5^	5~10 min × 3	5–10 min	Y	TESI ^6^ VAS ^7^
Frye et al. 2007	Yang style	60 min × 3	12 weeks	Y	STAI ^8^ SAS ^9^ CES-D ^10^
Gallegos et al. 2016	Tsung Hwa/Canneti/Rooting/Chen style (Mixed)	30 min × 2	30 × 2 min	Y	DASS-21 ^11^
Hong et al. 2017	N/A	60 min × 4	2 months	NR	SCL-90 ^12^
Li, A et al. 2006	N/A	30 min × 5	15 weeks	NR	SCL-90
Li, F et al. 2001	Condensed, classical Yang form	60 min × 2	24 weeks	Y	CESD-20 ^13^ PANAS-20 ^14^ PWPD ^15^ SWLS ^16^
Liao et al. 2018	Yang Style	50 min × 3	3 months	N/A	GDS ^17^
Noradechanunt et al. 2017	12 Movement Sun style	60 min × 2	12 weeks	N^2^	CES-D
Sattin et al. 2005	N/A	60~90 min × 2	48 weeks	Y	CES-D
Schitter et al. 2016	Yang style	60 min × 2	12 weeks	Y	CES-D ADS-K ^18^
Zhang et al. 2014	N/A	30~60 min × 3	18 months	Y	SECF ^19^ HAMA ^20^ HAMD ^21^
Zheng et al. 2017	Sim-24	60 min × 2	12 weeks	Y (First 6 weeks)	STAI PSS ^22^ SF-36 ^23^

^1^ Y: Yes; ^2^ N: No; ^3^ PHCSCS: Piers–Harris Children’s Self-Concept Scale; ^4^ BAI: The Beck Anxiety Inventory; ^5^ N/A: Not reported; ^6^ TESI: Tension and Effort Stress Inventory; ^7^ VAS: A bipolar visual analogue scale; ^8^ STAI: the State-Trait Anxiety Inventory; ^9^ SAS: Spielberger Anxiety Scale; ^10^ CES-D: the Center for Epidemiological Studies Depression Scale; ^11^ DASS-21: the Depression Anxiety Stress Scales; ^12^ SCL-90: Self-reporting Inventory-90; ^13^ CESD-20: 20-item Center for Epidemiologic Studies Depression scale; ^14^ PANAS-20: the 20-item Positive and Negative Affect Schedule; ^15^ PWPD: Positive well-being and psychological distress (subscale of the Subjective Exercise Experiences); ^16^ SWLS: the Satisfaction with life scale; ^17^ GDS: the Geriatric Depression Scale; ^18^ ADS-K: Allgemeine Depressionsskala -Kurzform questionnaire; ^19^ SECF: SECF Cognitive Scale; ^20^ HAMA: Hamilton Anxiety Scale; ^21^ HAMD: Hamilton Depression Scale; ^22^ PSS: The Perceived Stress Scale; ^23^ SF-36: 36-item short form survey.

**Table 3 ijerph-16-03033-t003:** Study quality assessment of all selected trials.

Studies	Item 1 ^1^	Item 2 ^2^	Item 3 ^3^	Item 4 ^4^	Item 5 ^5^	Item 6 ^6^	Item 7 ^7^	Item 8 ^8^	Item 9 ^9^	Score
Bao et al. 2015	0	1	0	1	0	1	1	1	1	6
Chang et al. 2013	1	0	0	1	0	1	1	1	1	6
Frith et al. 2011	0	0	0	0	0	1	1	1	1	4
Frye et al. 2007	1	1	0	1	0	1	1	1	1	7
Gallego et al. 2016	0	1	0	1	0	1	1	1	1	6
Hong et al. 2017	0	1	0	0	0	1	1	1	1	5
Li, A et al. 2006	0	1	0	0	0	1	1	1	1	5
Li, F et al. 2001	1	1	0	1	0	0	1	1	1	6
Liao et al. 2018	1	1	0	1	0	1	1	1	1	7
Noradechanunt et al. 2017	1	1	0	1	0	0	1	1	1	6
Sattin et al. 2005	1	1	0	1	1	1	1	1	1	8
Schitter et al. 2016	1	1	1	0	1	1	1	1	1	8
Zhang et al. 2014	1	0	0	1	0	0	1	1	1	5
Zheng et al. 2017	1	1	0	1	0	0	1	1	1	6

Item 1 = specified eligibility criteria; Item 2 = random allocation; Item 3 = concealed allocation; Item 4 = baseline equivalence; Item 5 = assessor blinding; Item 6 = retention rate ≥85%; Item 7 = intention-to-treat analysis for missing data; Item 8 = between-group statistical comparison; and Item 9 = point measure/measure of variability ≥ one key outcome.

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
