# Peer review of "The Effect of Tai Chi Chuan on Negative Emotions in Non-Clinical Populations: A Meta-Analysis and Systematic Review"

_ijerph, 2019, doi:10.3390/ijerph16173033_

Round 1

Reviewer 1 Report

Although not necessarily original, the paper reviews the outcome of a relatively simple therapy in the treatment of clinical conditions that deserve attention. The manuscript is presented with all available methodological rigor. The steps followed the academic format, and they are didactically described in good English. The findings may support short-term actions in public health initiatives.

A good differential of this paper was the search in English and Chinese language databases. This is particularly important for audiences who cannot access papers in the original language of this technique. The limitations of the study were well stated in the Discussion. However, there is still some doubt about the validity of comparing results from studies that used different TCC modalities (styles). Given the complexity of prevention and treatment of anxiety and depression, the phrase " an efficient preventive tool" (line 315, and also in Abstract) would be better expressed as "a worthy complementary non-pharmacological resource". Lastly, something that could also be added at the text closure would be what is expected of future studies, to make a link with the continuing process of knowledge.

Reviewer 2 Report

This paragraph concerns me a bit:

"There is a wide range of evidence-based preventive strategies available, which have been found to reduce risk factors and strengthen protective factors relating to emotional disorders. Among these preventive approaches, physical exercise, such as aerobic exercise and Tai Chi Chuan (TCC), could provide both physical and psychological benefits [7]."

I have watched several TCC-tutorial videos due to I do not pretty much about TCC. My concern is how this practice can be assimilated as an aerobic exercise like other aerobic-physical activities such as running or trekking. To what extent TCC provides something different from other aerobic physical activities? 

Before I endorse the paper, I'd recommend to include a paragraph about Taoism philosophy. Obviously, TCC supplies something more than just an aerobic activity. On the other hand, the article accomplishes the minimum standards of a systematic review. 
